# Estimating Long-Run Relationship between Renewable Energy Use and CO₂ Emissions: A Radial Basis Function Neural Network (RBFNN) Approach

**Pradyot Ranjan Jena** [1,*] , **Babita Majhi** [2] and **Ritanjali Majhi** [1]

1 School of Management, National Institute of Technology Karnataka, Surathkal, Mangalore 575025, India; ritanjali@nitk.edu.in
2 Department of CSIT, Guru Ghasidas Vishwavidyalaya (Central University), Bilaspur 495009, India; babita.majhi@gmail.com
* Correspondence: pradyotjena@nitk.edu.in; Tel.: +91-824-247-3237

**Abstract:** The long-run relationship between economic growth and environmental quality has been estimated within the framework of the environmental Kuznets Curve (EKC). Several studies have estimated this relationship by using statistical models such as panel regression and time series regression. The current study argues that there is a nonlinear relationship between environmental quality indicators and economic and non-economic predictors and hence an appropriate nonlinear model is required to predict it. An adaptive and nonlinear model, namely radial basis function neural network (RBFNN) has been developed in this study. $CO_2$ emission is used as the target output and renewable energy consumption share, real GDP, trade openness, urban population ratio, and democracy index are used as the predictors to estimate the EKC relationship for nineteen major $CO_2$ emitting countries that account for 78% of the global emissions. The model developed in this study could predict the $CO_2$ emissions of all the countries with more than 95% accuracy. This finding underlines the usefulness of the RBFNN model which can be used to predict emission levels of other pollution indicators at the global level. Further, comparing two models, one with all the predictors and the other excluding the renewable energy share, it was found that the model with renewable energy share predicts $CO_2$ emissions more accurately. This reinforces the already strengthening campaign to encourage industries and governments to increase the share of renewable energy in total energy use.

**Keywords:** EKC estimation; $CO_2$ emissions prediction; neural networks; radial basis function neural network; renewable energy consumption

## 1. Introduction

The likely impacts of economic growth on environmental degradation have been analyzed and examined by economists for decades now but there is still no consensus on how different predictors such as trade openness and energy consumption affect environmental degradation [1]. Recent studies have highlighted the contribution of non-economic factors such as democracy in determining the environmental quality of a country [2,3]. A lack of consensus can be attributed to the countries studied, the period chosen, the choice of explanatory variables, and the methodologies used. The pioneering studies by the early researchers such as Grossman and Krueger [4,5], Shafik and Bandyopadhyay [6], and Selden and Song [7], have been continued with significant contributions by the later researchers over the years and produced a large number of empirical studies, which has popularly come to be known as "environmental Kuznets curve" (EKC). An inverted U-shaped EKC hypothesis states that as a country's economy develops, environmental pollution increases initially and then begins to decline until it reaches a certain income level

threshold. Once a certain (threshold) income level is attained, this results in an environmental improvement [5,8,9]. Antweiler et al. [10] broke down the influence of international trade on the environment into three distinct effects: scale, composition, and technique, and then summed them together to calculate the overall impact of free trade on environmental quality. Later, Managi et al. [11], Tsurumi and Managi [12,13], Kagohashi et al. [14], and Abe et al. [15] produced more realistic results in the EKC relationship by treating income and trade openness as endogenous variables.

Although numerous studies produce different estimates of EKC, there is still a common shortcoming in these studies. The methods used are either time-series causality and cointegration tests or panel regressions and panel cointegration regressions. These methods typically estimate a single constant parameter for the relationship for the entire sample period. Even though some prominent research takes into account structural breaks in their estimated EKC relationship, they still produce constant estimates of the effect of economic growth on indicators of environmental quality over the entire predicted period [16]. We argue that there is a potential nonlinear relationship between air pollution and its economic predictors such as GDP per capita, renewable energy consumption, and trade openness over a period of time. If the apparent nonlinearities existing in this relationship are explicitly modeled, more accurate predictions can be made. This is the major contribution of this study to the EKC literature. We develop a nonlinear dynamic neural network model, namely the radial basis function neural network (RBFNN) model to predict the $CO_2$ emissions of 19 countries based on the economic factors such as real GDP (constant US$), renewable energy share in total energy use, and trade openness measured by export and import ratio to GDP and non-economic factors such as democracy status of a country and urban population ratio. In the RBFNN model, the predictors (inputs) are passed through a Gaussian function to receive information from each other through nodes (neurons) that enhance their prediction ability. The adjoining weights are continuously adjusted by the adaptive error learning process and the final output ($CO_2$ emission) is produced.

The other major contribution of this study is to highlight the effect of renewable energy consumption on the emission path of $CO_2$. Though several studies have used this variable in EKC estimation as detailed in Section 2, none of them have measured the accuracy of their estimations. These studies in the linear statistical framework estimated a single constant parameter for renewable energy's effect on environmental quality indicators. But whether these estimates could reliably predict the $CO_2$ emission path for the entire sample period they used is questionable. Unless, the studies compared the similarity between the predicted and actual level of emissions based on their estimated parameters and found a higher level of similarity, the validity of the estimates is doubtful. On this premise, we compared the predictive accuracy of our model by comparing the actual and predicted figures of $CO_2$ using the mean absolute percentage error (MAPE) values and found a very small error percentage. Furthermore, we used two specifications to predict $CO_2$ emissions for all countries. In the first specification, all the inputs except for renewable energy share are used as inputs and in the second, the latter is added to the list of inputs. Then, we compared the MAPE of the two specifications and found out that the MAPE of the specification in which renewable energy is used is much smaller for most countries compared to the one in which it is not used. This comparison of model predictions validates the contribution of renewable energy in reducing $CO_2$ emissions beyond a reasonable doubt.

We have used only one environmental indicator in this study i.e., $CO_2$ emission as this is considered the biggest contributor to climate change and has been given special attention in the reports of the Intergovernmental Panel on Climate Change (IPCC).

Finally, the democratic status of a country has been used as a non-economic factor in the non-linear neural network model. Only a very few studies have used this indicator to determine the shape of the EKC but they used it in the linear regression framework [2,3]. The nineteen countries selected for this study are the major emitters of $CO_2$. Eleven of these countries emit either 2% or more of the total global emissions and the rest eight countries

emit 1%. They together account for 78% of the global $CO_2$ emissions. The details of the variables used and the source of the data are provided in Section 3. The RBFNN model is explained in Section 4. The simulation procedure is described in Section 5 and the results are interpreted in Section 6. Finally, Section 7 concludes with policy implications.

## 2. Literature Review

In recent years, the role of renewable energy consumption in the EKC relationship has been examined by various authors and the relationship between renewable energy and $CO_2$ emissions was found to be less clear-cut. While Sugiawan and Managi [17], Sinha and Shahbaz [18], Liu et al. [19], and Apergis et al. [20] claim that increasing renewable energy consumption will result in a long-run reduction in $CO_2$ emissions, other studies such as Adams and Nsiah [21], Saidi and Omri [22] found that renewable energy increases $CO_2$ emission in some countries while reducing in some others. A few other studies such as Menyah and Wolde-Rufael [23], Sinha et al. [24], and Tanti et al. [25] have found no significant long-term relationship between renewable energy consumption and $CO_2$ emission. Liu [26] while reviewing China's renewable energy law and policy observed several hindrances to higher use of renewable energy, such as problems with fragmentation, obsolescence, and lack of operability. Chen et al. [27] examined the possibility of an EKC relationship using provincial data in China spanning a period from 1995 to 2012. Their results show a heterogenous effect wherein there is no evidence of an inverted U-shaped relationship in the central and western regions but was observed in the eastern region.

Bilgili et al. [28] using a dataset for a period spanning 2003–2018 on a set of developed countries, discovered an EKC relationship only for higher $CO_2$ emitting countries. The N-shaped nexus, on the other hand, is more prevalent in countries with lower carbon emissions. They also discovered that research and development in energy efficiency is more effective at reducing carbon emissions than research and development in fossil fuels and renewable energy sources combined. Gyamfi et al. [29] by using data from 1995 to 2018, found no evidence of an N-shaped EKC in the countries under study; instead, they found an inverted U-shaped EKC relationship. They recommended that the usage of renewable energy be increased to reduce pollution emissions in these countries. Kirikkaleli and Adebayo [30] based on data for the period 1990–2015 and different time series econometric models found a long-run relationship between $CO_2$ emissions and their probable drivers. They discovered that long-term public-private partnership investment in energy has a favorable impact on $CO_2$ emissions. Yang et al. [31] using a dataset of manufacturing industries from 38 countries observed that increased consumption of renewable energy has resulted in modifications in the relationship between manufacturing growth and $CO_2$ emissions. Using data from the BRICS economies over a period from 1980 to 2016, Khattak et al. [32] examined the role of technological innovation and renewable energy use in the $CO_2$ emissions growth path. They discovered that except for Brazil, innovative efforts failed to reduce $CO_2$ emissions in China, India, Russia, and South Africa. They also demonstrated that except for South Africa, the increase in renewable energy use has helped reduce $CO_2$ emissions in the BRICS panel.

Using data from 31 provinces of China between 2007 and 2017, Zeraibi et al. [33] found that government expenditure has a positive effect on environmental quality in China. Chen et al. [34] using the panel data from China from 1980 to 2014, found a long-run relationship between per capita $CO_2$ emissions and the economic predictors. They discovered that economic growth, non-renewable energy generation, and international trade do not show an EKC relationship with $CO_2$ emissions but the inclusion of renewable energy production in the inputs confirmed the U-shaped EKC hypothesis. Khan et al. [35] using data from 34 high-income countries over the period 1995–2017 show a reciprocal relationship between GHG emissions and renewable energy in 22 countries. Yao et al. [36] using a dataset of 17 developing and developed countries spanning a period from 1990 to 2014, found the existence of both the EKC and renewable energy Kuznets Curve (RKC)

hypotheses. They showed that a 10% increase in renewable energy consumption rate led to a reduction in carbon emissions by 1.6%.

Zeraibi et al. [37] used the levels of government expenditure as fiscal and broad money supply as monetary policy instruments to predict $CO_2$ emissions. Their findings reveal that expansionary fiscal policy led to an increase in $CO_2$ emissions whereas expansionary monetary policy decreased it in both the short- and long-run in China. They could not find evidence for the EKC hypothesis, rather the relationship between economic growth and carbon emissions was N-shaped. A carbon emission function was used by Balsalobre-Lorente et al. [38] to examine an EKC relationship between economic growth and $CO_2$ emissions in five European Union countries for the period 1985 to 2016. In the EU-5 countries, they discovered an N-shaped association between economic growth and $CO_2$ emissions. Furthermore, they discovered that the use of renewable electricity, the use of natural resources, and the use of innovative energy technologies all contribute to improved environmental quality. Using panel data from G20 countries, it has been shown by Paramati et al. [39] that FDI inflows reduce $CO_2$ emissions both in developed and developing economies, but stock market expansion slows in developed economies. They also discovered that the use of renewable energy significantly cuts $CO_2$ emissions while simultaneously increasing economic production across the countries represented in their panels. After conducting research on 30 nations over the period 2000 to 2013, Kim and Park [40] concluded that developing the financial sector in a country can aid in the deployment of more renewable energy, which in turn can assist reduce $CO_2$ emissions.

Apart from the economic factors, the environmental quality may also be affected by the non-economic factors such as the political institutions that are involved in the process of environmental policymaking in a country [41]. Several environmental problems, according to Romuald [42], can be attributed to institutional failure and ineffective government practices and policies. Goel et al. [43] claim that numerous measures have been enacted to compel economic agents to internalize environmental externalities (directly or indirectly). A critical aspect in the success of these initiatives is a country's institutional quality. Within this body of literature, some scholars have concentrated on the democracy–pollution nexus, while others have evaluated the effect of political freedom on pollution.

A few studies have taken into account political variables that are related to the income–pollution relationship [44,45]. The findings are mixed when examined empirically. According to the findings of the studies by Torras and Boyce [45], Barrett and Graddy [44], Li and Reuveny [46], and Farzin and Bond [47], democratization results in citizens being better informed and better equipped to demonstrate their dissatisfaction with government. Torras and Boyce [45] discovered that democracy had a favorable and statistically significant impact on environmental quality in general, and particularly in low-income nations. Farzin and Bond [47] discover evidence suggesting a country's level of democracy and the liberties that come with it are positively related to the condition of the environment. Several academics, on the other hand, believe that democracy may not improve or even deteriorate environmental quality [48–50]). Roberts and Parks [49], for example, conclude that democracy does not affect carbon emissions. In addition, Scruggs [50] finds that when wealth disparity is taken into account, there is no significant association between democracy level and three environmental indicators (dissolved oxygen demand, fecal coliform, and particle emissions). Midlarsky [48], on the other hand, indicates that a higher level of democracy is connected with a worse environmental performance in a country.

## 3. Materials

The International Energy Agency (IEA, Paris, France) has compiled data on carbon dioxide ($CO_2$) emissions from the combustion of natural gas, coal, oil, and other fuels, as well as emissions from industrial waste and nonrenewable municipal waste. This data has been used to select 19 countries based on their emission intensity as shown in Table 1. The website from which the emission shares are reproduced is "Each Country's Share of $CO_2$ Emissions | Union of Concerned Scientists (ucsusa.org)". The top emitting countries

whose share is more than 2% of the global emission are China, U.S., India, Russia, Japan, Iran, South Korea, Saudi Arabia, Indonesia, Germany, and Canada. The rest eight countries considered in this study have a share of 1%.

**Table 1.** Fossil $CO_2$ emissions share and the absolute values of $CO_2$ emissions for selected countries.

| Sl. No. | Emission Share of Selected Countries |
|:---:|:---:|
| 1 | China (28%) |
| 2 | U.S. (15%) |
| 3 | India (7%) |
| 4 | Russia (5%) |
| 5 | Japan (3%) |
| 6 | Iran (2%) |
| 7 | South Korea (2%) |
| 8 | Saudi Arabia (2%) |
| 9 | Indonesia (2%) |
| 10 | Germany (2%) |
| 11 | Canada (2%) |
| 12 | Brazil (1%) |
| 13 | South Africa (1%) |
| 14 | Mexico (1%) |
| 15 | Turkey (1%) |
| 16 | Australia (1%) |
| 17 | United Kingdom (1%) |
| 18 | Italy (1%) |
| 19 | France (1%) |

The data on the predicted variable i.e., $CO_2$ emissions, and the predictors such as GDP in constant US$ measured in 2010, renewable energy share in total energy use, the urban population as a percentage of the total population, and trade openness for all 19 countries are drawn from the World Bank database for the period 1960 to 2019. The data for another predictor i.e., democracy is obtained from the database of Freedom House, which is an independent watchdog organization based in the USA. It collects and publishes data on the political rights (PR) and civil liberties (CL) of most countries of the world. The democracy index used in this study is constructed by adding the scores of PR and CL of the nineteen countries. The description of output and input variables and the data sources are provided in Table 2. The data files are available in the Supplementary Materials section of this article.

**Table 2.** Variable description and data source.

| Variables | Data Source |
|:---:|:---:|
| Carbon dioxide emissions (mega ton) | World Development Indicators [51] |
| Renewable energy share in total energy use (%) | World Development Indicators [51] |
| GDP (constant 2005 US$) | World Development Indicators [51] |
| Urban Population Ratio | World Development Indicators [51] |
| Trade openness (ratio of imports plus exports to GDP | World Development Indicators [51] |
| Sum of the Freedom House Political Rights and Civil Liberties Indices | Freedom House [52] |

Notes: All the data are annually from 1960 to 2019. Freedom in the World | Freedom House. http://data.worldbank.org/indicator. Accessed on 2 February 2022.

The compound annual growth rate (CAGR) of $CO_2$ emissions of the 19 countries between 1990 and 2019 is shown in Figure 1. The countries that experienced higher levels of $CO_2$ emissions during this period are China, India, Saudi Arabia, Iran, Turkey, and Brazil with 6.7%, 6.2%, 4.8%, 5.3%, 4.6%, and 3.2% respectively. On the other hand, the UK with −1.8%, Italy with −0.9%, France with −0.58%, the USA with 0.11%, Japan with 0.05%, and Canada with 1.3% are the countries that have managed a low emission growth path. The

trends in Figure 1 indicate the existence of an EKC relationship as the $CO_2$ emissions have declined in developed countries and increased in highly developing countries.

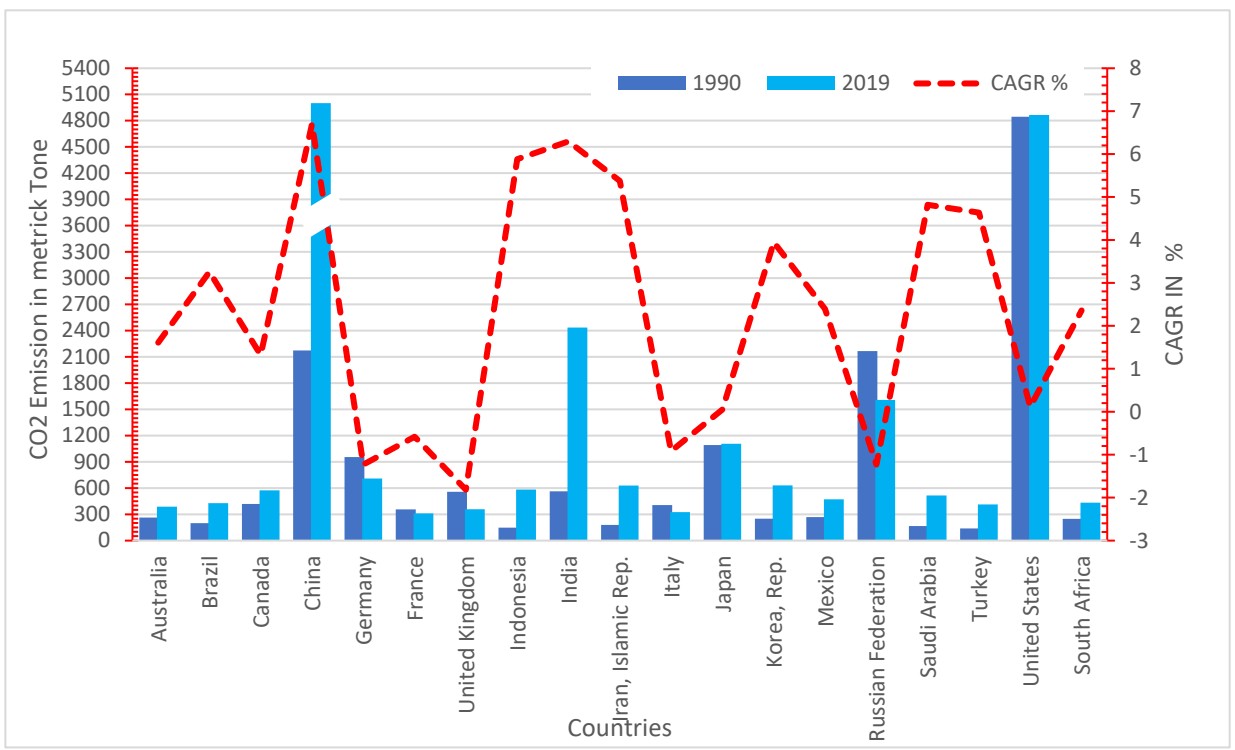

**Figure 1.** The growth rate of $CO_2$ emission (mt).

## 4. Development of Radial Basis Function Neural Network (RBFNN) Based $CO_2$ Prediction Model

Artificial neural networks (ANN) are nonlinear models having a lot of real-life applications. There are different types of architecture available under ANN such as feed-forward networks, and feedback networks which might be single layer or multilayer. Depending upon the nonlinearity associated with the problem the network is chosen judiciously. The RBFNN is a simple single hidden layer feed-forward network trained by a supervised learning algorithm [53]. The hidden layer nodes also known as centers use radial basis functions (RBF) or Gaussian functions. The nonlinear mapping of the data from the input to the output layer is done as it passes through the RBF or Gaussian functions. Mathematically, the RBF calculates the Euclidean distance between the input data and the nodes or centers present in the hidden layer. The weighted sum of the output of RBF nodes is considered the final output of the network.

The advantages of the RBFNN model in the prediction process are as follows:

(1) Training is faster in RBFNN as it involves a smaller number of computations. Hence it gives faster convergence.
(2) The function of each hidden node can be easily interpreted in RBFNN.
(3) There is no requirement to decide apriori the number of hidden layers in RBFNN, which is needed in some other models.

Taking into consideration the above advantages, the RBFNN model is used for the development of $CO_2$ emission prediction which is an optimization problem.

The block diagram of RBFNN based prediction model is shown in Figure 2. Each node in the hidden layer is an RBF or Gaussian function having a center and width. Let the centers and corresponding widths associated with $h$ number of nodes in the hidden layer be represented as $c = c_1, c_2, c_3 \ldots c_h$ and $\sigma = \sigma_1, \sigma_2, \sigma_3 \ldots \sigma_h$ respectively. The same input $(x = x_1, x_2, x_3 \ldots x_n)$ is given to all the nodes of the hidden layer. The dimension of centers

of every hidden node and the input data are the same, i.e., $c_i \in R^n, x \in R^n$. The output of each hidden node $(\phi_1, \phi_2, \phi_3 \ldots \phi_h)$ is multiplied by the weight values $(w_1, w_2, w_3 \ldots w_h)$ respectively to produce the final output of the network.

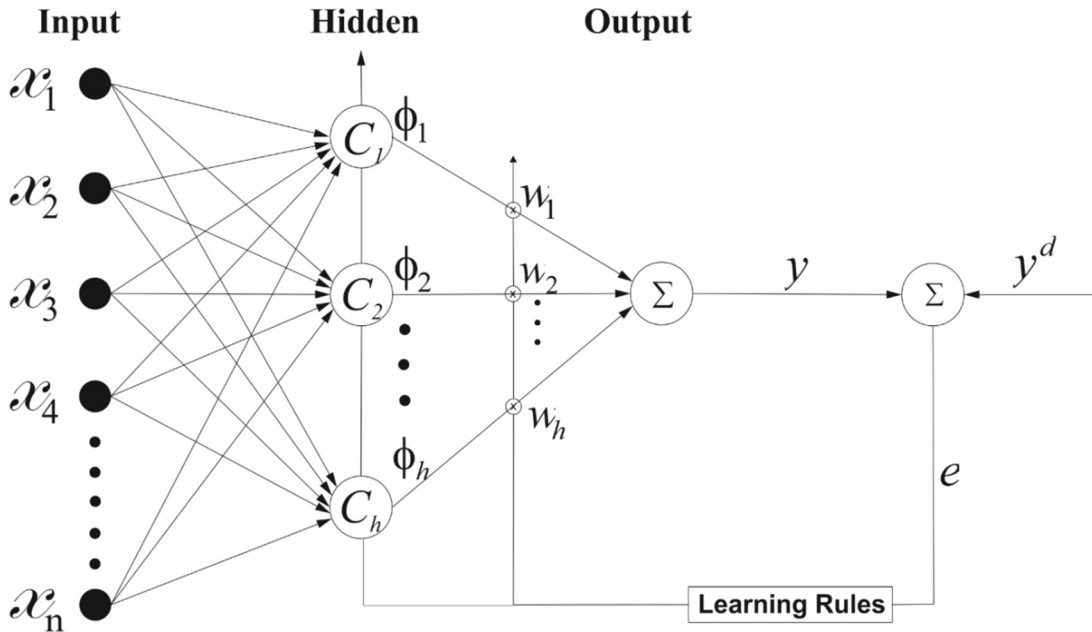

**Figure 2.** A schematic diagram of RBFNN based $CO_2$ prediction model with n number of inputs.

The output of $i$th hidden node $\phi_i$ is represented as

$$\phi_i(z) = e^{\frac{-z^2}{2\sigma_i^2}} \tag{1}$$

where, $z = ||x - c_i||$, denotes the Euclidean distance between input data and the corresponding centers and $\phi_i = \phi(||x - c_i||)$. The final response of the RBFNN for a particular input is calculated as

$$y = \sum_{i=1}^{h} w_i \phi_i \tag{2}$$

Training of the RBFNN model is carried out iteratively for each training data, $\{x, y\}$. During this learning period the model parameters such as the weights, centers, and width values, $\{w_i, c_i, \sigma_i\}$ are updated until the error cost function is minimized. The error cost function $e$ is given as

$$e = \frac{1}{2}\left(y^d - y\right)^2 \tag{3}$$

At any time instant $t$, the parameter update rules to change $\{w_i, c_i, \sigma_i\}$ are given below. The update rules are derived using the gradient descent algorithm.

$$w_i(t+1) = w_i(t) + \eta_1 \left(y^d - y\right)\phi_i \tag{4}$$

$$c_{ij}(t+1) = c_{ij}(t) + \frac{\eta_2}{\sigma_i^2}\left(y^d - y\right)w_i\phi_i\left(x_j - c_{ij}\right) \tag{5}$$

$$\sigma_i(t+1) = \sigma_i(t) + \frac{\eta_3}{\sigma_i^3}\left(y^d - y\right)w_i\phi_i z_i^2 \tag{6}$$

where, $y^d$ = desired or target value. In this case, it is the $CO_2$ emission value.

$c_{ij}$ = $j$th element of $i$th center.

$\eta_1$, $\eta_2$, $\eta_3$ = learning rate for network parameters, $\{w_i, c_i, \sigma_i\}$ respectively.

## 5. Simulation Study

The simulation procedure explains the steps that are carried out during the development of the RBFNN based $CO_2$ emission prediction model. The three main steps involved in it are data preprocessing, training, and testing of the model.

### 5.1. Data Preprocessing

The data is collected from 19 different countries from 1960 to 2019. The EKC relationship is estimated using the $CO_2$ emissions as a parameter for environmental quality, renewable energy share in total energy used, the urban population as a percentage of the total population, real GDP, trade openness, and political freedom as the predictors of $CO_2$ emissions. The main objective of this study is to predict the $CO_2$ emission levels of major emitting countries based on the key predictors and to highlight the role of renewable energy in predicting $CO_2$ emission. For the second objective, we have used two specifications of the model. In the first specification, renewable energy share is excluded (partial model) and in the second all the predictors are used (full model). The purpose is to compare the predictive performance of the full model against the partial model. The hypothesis here is that the performance of the full model will be higher than the partial model, which would entail renewable energy as the major predictor of $CO_2$ emission. In the RBFNN model developed in the study, $CO_2$ emission is taken as the target output and the predictor variables as the inputs. The data for the target and input variables are normalized before they are used to develop the model. Normalization of the data is done by dividing each value of each column by the corresponding maximum value. Hence all the values lie between 0 to 1. Normalization is one of the important steps of data preprocessing as the RBFNN model is used for prediction purposes. The normalization of the data helps in faster convergence of the model. After normalization, the dataset is divided into two sets–training and testing sets. Randomly selected 80% of the data becomes the training set which is used to develop the RBFNN model and the remaining 20% of data becomes the testing set which is used for the evaluation of the model. As the sample size for each country contains 59 data tuples, randomly 47 data tuples (80%) are selected for the training of the model and 13tuples for the testing.

### 5.2. Training of the Model

During the training process, the neural network model learns from the past data iteratively and becomes adaptive. Referring to Figure 3, the RBFNN structure used for the simulation is 5:4:1. It has five inputs, four nodes or centers in the hidden layer, and one output. The four nodes of the hidden layer contain Gaussian functions. Each Gaussian function has a center and center-width. The number of centers at each Gaussian function is equal to the number of inputs. Since the number of inputs is five, in this case, each of the Gaussian functions at each neuron has five centers. Initially, the value of centers, center-width of Gaussian functions, and the connecting weights are initialized to remain between −0.5 to +0.5. Out of the training data set, a single data point containing five values is given as input to the model. It is then passed through the Gaussian functions of the hidden layer, multiplied with the corresponding weight values, and summed over to produce the estimated output. The error value is obtained by comparing the estimated output with the corresponding target value. The error value may be a positive or negative, hence squared error which is always positive is used as the cost function which needs to be minimized. Using the error value and the learning algorithm of RBFNN the weights, centers, and widths are updated. The detailed update equations are given in Equations (4)–(6). The process is repeated for all inputs and the corresponding error square values are calculated. This completes one experiment. This simulation process is repeated 2000 times until the mean squared error is minimized. The mean square error (MSE) value for each experiment or iteration is noted and plotted against the iteration to observe the convergence characteristics. The details of the parameters used for simulation are given in Table 3. Once the MSE is

minimized the final value of weights, centers, and center-width are frozen. The model is then ready for testing purposes.

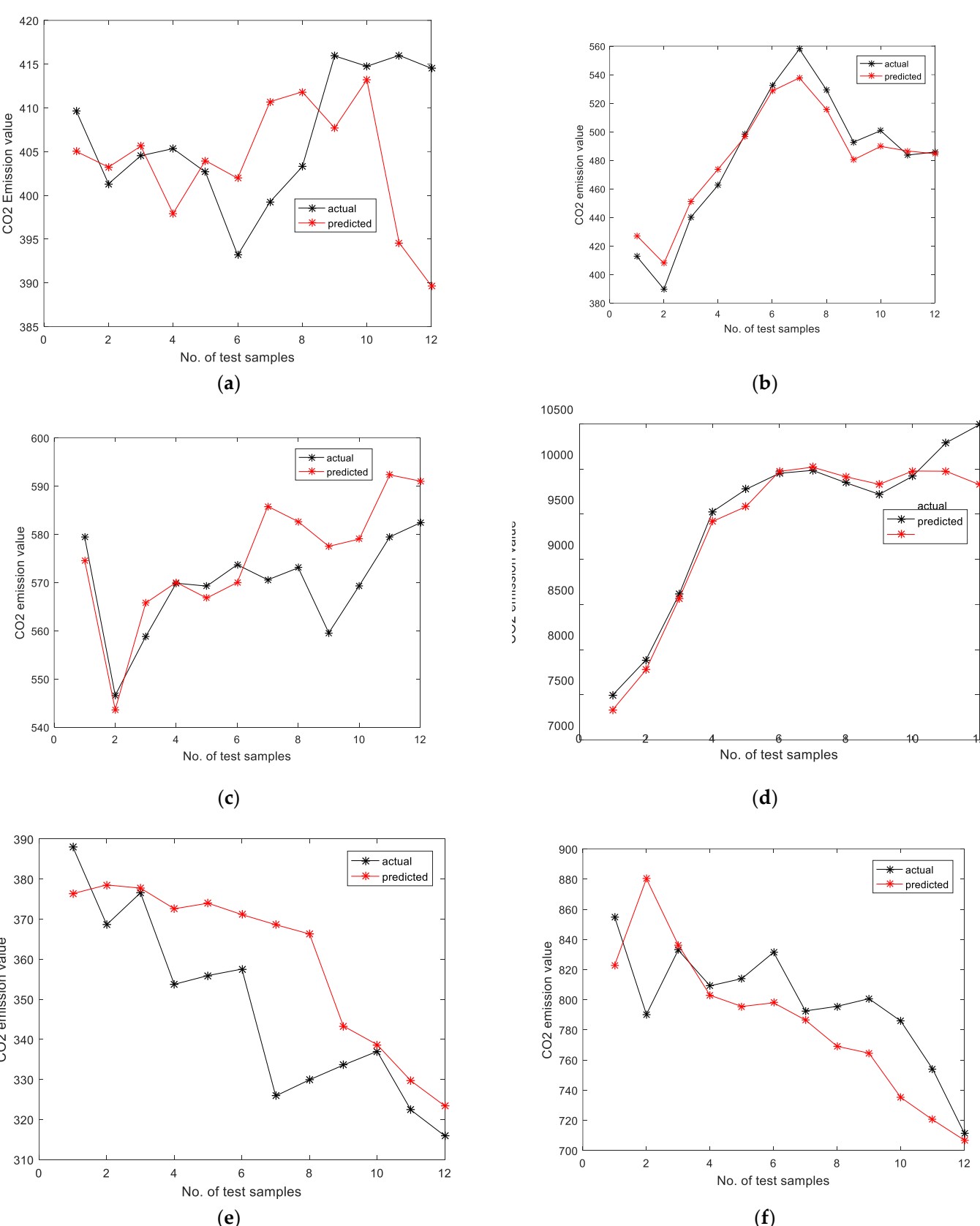

**Figure 3.** *Cont.*

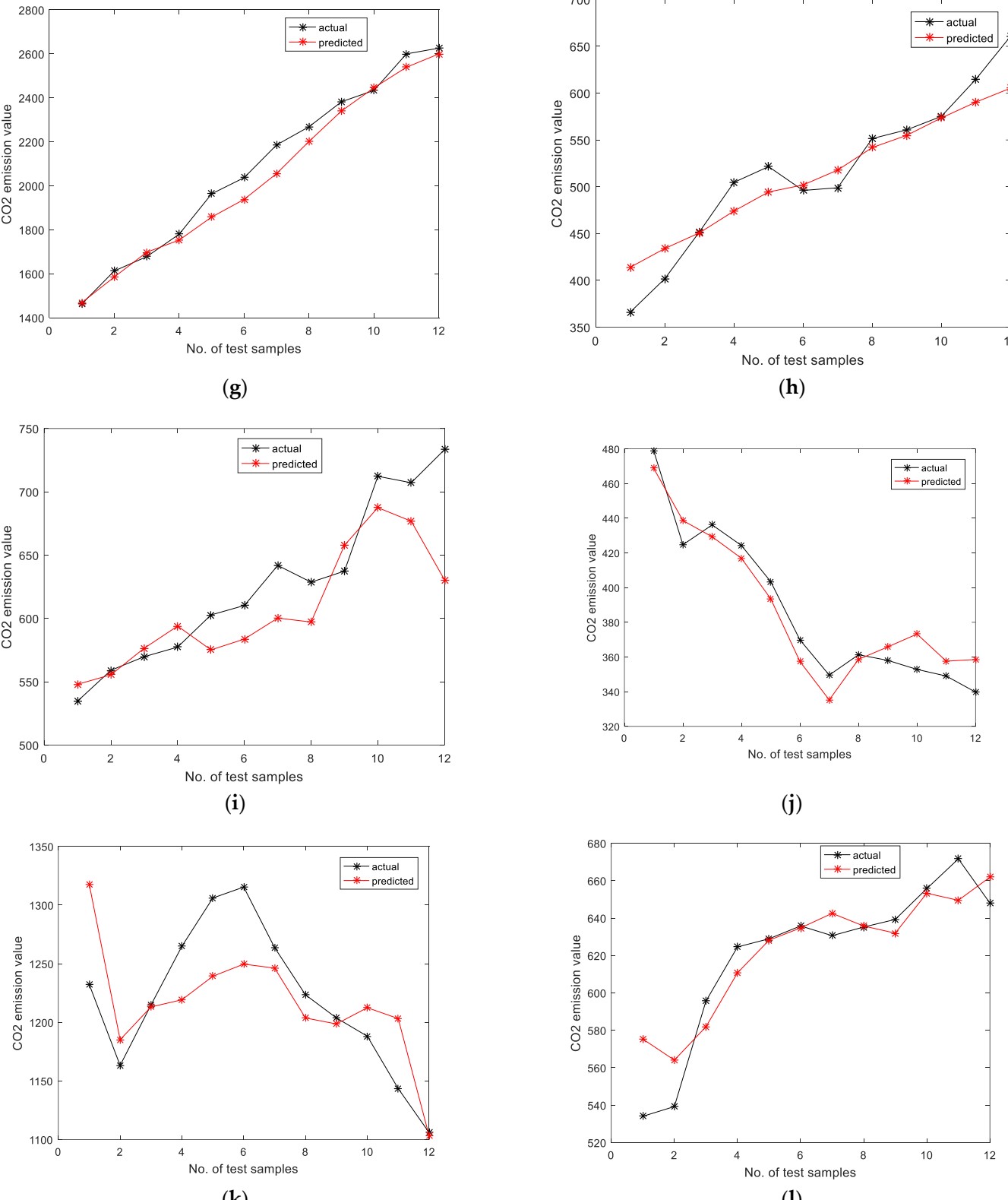

**Figure 3.** *Cont.*

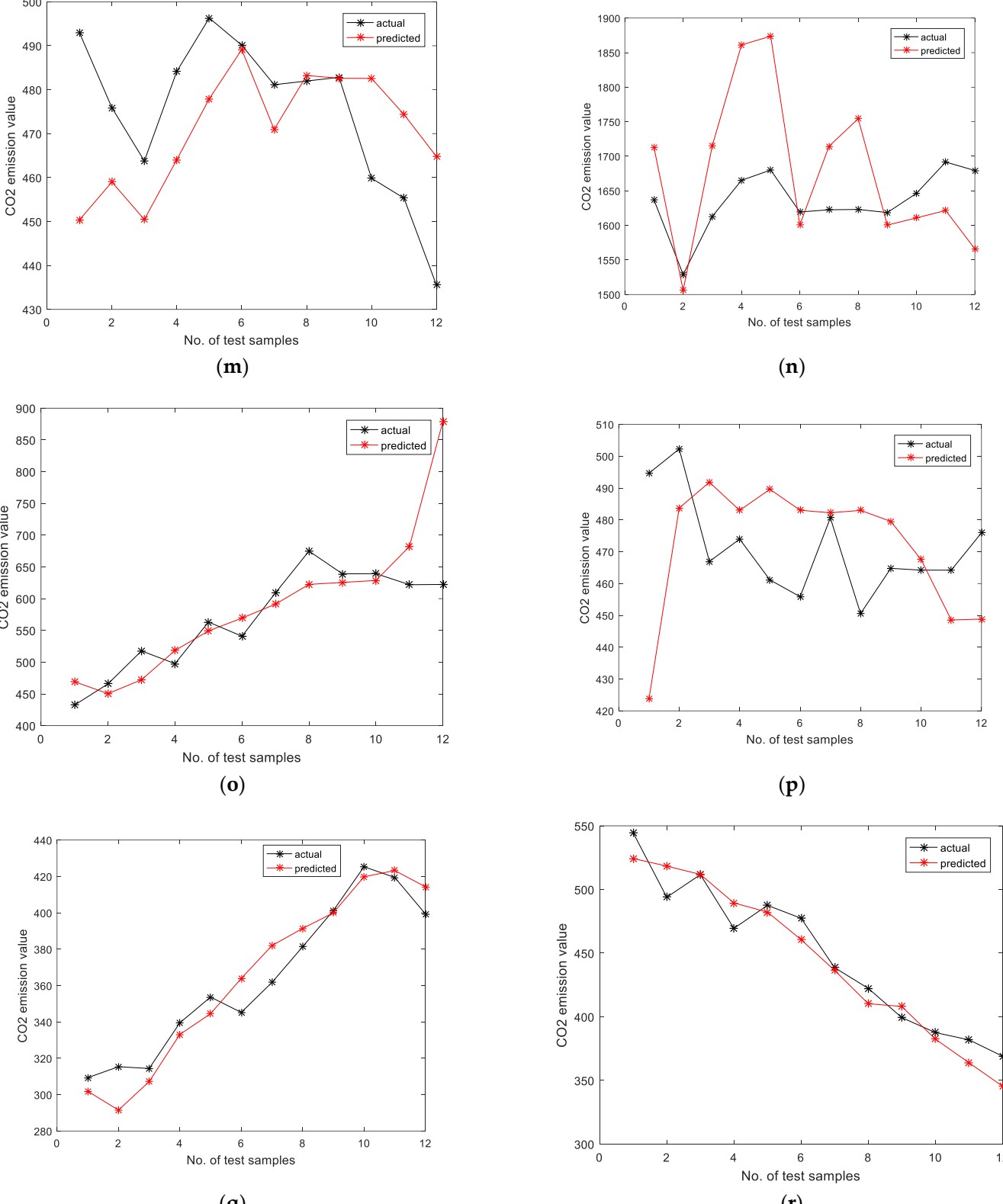

**Figure 3.** *Cont.*

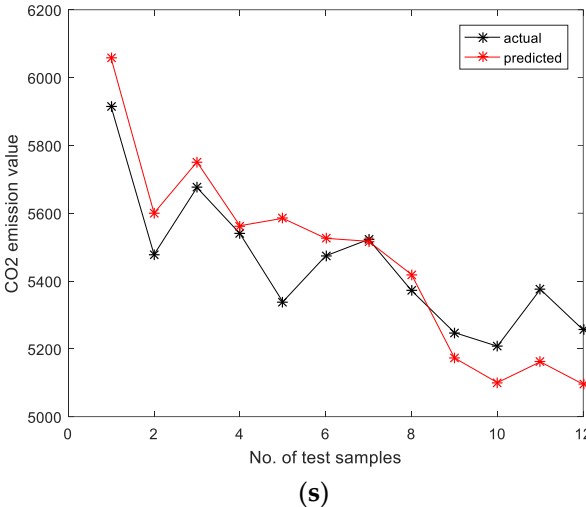

**(s)**

**Figure 3.** Actual and estimated $CO_2$ emission values during testing using the RBFNN model (**a**) for Australia; (**b**) for Brazil; (**c**) for Canada; (**d**) for China; (**e**) for France; (**f**) for Germany; (**g**) for India; (**h**) for Indonesia; (**i**) for Iran; (**j**) for Italy; (**k**) for Japan; (**l**) for the Korea Republic; (**m**) for Mexico; (**n**) for Russia; (**o**) for Saudi Arabia; (**p**) for South Africa; (**q**) for Turkey; (**r**) for the UK; (**s**) for the USA.

**Table 3.** Parameters used in the simulation.

| Parameter | Value |
|---|---|
| Structure of RBF full model | 5:4:1<br>(No. of inputs: 5, hidden neurons: 4, output: 1) |
| Structure of RBF partial model | 4:4:1<br>(No. of inputs: 4, hidden neurons: 4, output: 1) |
| Number of Centres or nodes in the hidden layer | 04 |
| Number of experiments | 2000 |
| Number of training tuples (80%) | 30 |
| Number of testing tuples (20%) | 07 |
| Value of μ (learning parameter) | 0.1 |

### 5.3. Testing of the Model

Once the model is trained, it is said to have been learned from the past data in an adaptive manner using an error correction method and well designed. After this, the model is being tested using the testing dataset to assess its prediction accuracy. Each data point of the testing set is used as an input to the model. These inputs are applied to the optimized RBFNN model, passed through the Gaussian function, weighted and then summed over to produce the estimated output of $CO_2$ emission value. Each of these estimated values is compared with the actual target value to evaluate the performance of the RBFNN based prediction model. The Mean absolute percentage error (MAPE) is calculated using Equation (7).

$$MAPE = \frac{1}{N}\sum_{l=1}^{N} abs((y^d(n) - y(n))/y^d(n) \times 100 \qquad (7)$$

where $N$ = no. of testing tuples.
$y^d(n)$ = desired value for the nth testing tuple.
$y(n)$ = the estimated value for the nth testing tuple.

## 6. Results

In this study, two models such as the full model (with renewable energy) and the partial model (without renewable energy) are used to compare the performance of prediction accuracy (Table 4, columns 3 and 4). The MAPE values in Table 4, Col. 3 exhibit that the RBFNN based prediction model can predict the $CO_2$ emission figures accurately as the MAPE is less than 5% for all the countries except for Russia and Saudi Arabia, which have 5.4% and 8.2% respectively.

**Table 4.** MAPE value for $CO_2$ emission prediction.

| Emission Intensity | Countries | Full Model (with Renewable Energy) | Partial Model (without Renewable Energy) |
|---|---|---|---|
| High-emission countries | China | 1.63 | 100.00 |
| | The USA | 1.95 | 6.44 |
| | India | 2.46 | 3.06 |
| | Russia | 5.40 | 100.00 |
| | Japan | 2.80 | 4.19 |
| | Iran | 4.38 | 4.76 |
| | South Korea | 2.17 | 2.78 |
| | Saudi Arabia | 8.17 | 4.98 |
| | Indonesia | 4.41 | 4.57 |
| | Germany | 3.56 | 5.56 |
| | Canada | 1.4 | 1.01 |
| Low-emission countries | Brazil | 2.16 | 4.65 |
| | South Africa | 4.82 | 6.47 |
| | Mexico | 3.45 | 5.32 |
| | Turkey | 3.05 | 6.73 |
| | Australia | 2.06 | 1.82 |
| | UK | 2.96 | 4.88 |
| | Italy | 2.94 | 11.38 |
| | France | 4.37 | 8.26 |

The linear regression models produce a single parameter estimate for the entire sample period. Hence, there is no adaptive process using the error to update the coefficients of the linear model. These linear models, therefore, produce a large error that makes the parameter estimates less precise. In contrast, the RBFNN model has an adaptive process that makes the model learn from the error iteratively and thus, helps in reducing the error with each iteration. This process of error learning through the feed-forward procedure makes the model adaptive. When the error is minimized completely, the final parameters are frozen. The weights can be interpreted as impact coefficients of the inputs with respect to the output variable, i.e., $CO_2$ emissions. Unlike the linear regression models, these coefficient values are not a single estimate, but rather produced through an adaptive error learning procedure and hence, yield highly precise parameter estimates. Along with the weights, the RBFNN model also produces optimal center values and the values of width.

From the 19 countries considered in this study, 11 are categorized as high emitting countries, each having a share of 2% or more. The rest 8 countries have a share of 1% each and are categorized as low emitting countries. We compared the MAPE values in the full model (Col. 3) with that of the partial model (Col. 4). The purpose is to show the relative contribution of renewable energy share in total energy used in the prediction of $CO_2$ emissions. Although some of the past studies have shown rather a strong effect of renewable energy in the EKC shape [34], given that they have used linear statistical models, the magnitude of the effect that they show may not be reliable. In this study, the RBFNN model provides a reliable prediction of $CO_2$ emissions, and hence, the difference in prediction accuracy between the full and partial models can be directly attributed to the renewable energy share. The full model has yielded less MAPE value for 17 countries out of the total 19, thus confirming the significant contribution of renewable energy share in

total energy in predicting the $CO_2$ emission value. The prediction accuracy of these country cases is nearly 98%.

The actual and estimated values of $CO_2$ obtained from the RBFNN model during testing are plotted in Figure 3a–s. The Figures show that there is a higher degree of convergence between the actual and estimated values of $CO_2$ during the testing.

China is the biggest emitter of $CO_2$ accounting for 28% of the global emissions. In the last decade, China has transformed its manufacturing sector to integrate the circular economy model that focuses on the reuse and recycling of materials. The country has set up industrial parks in which the principles of the circular economy have been integrated into the entire supply chain of the companies [54]. Despite these efforts, China is expected to remain the biggest emitter of $CO_2$ with a rising share of the emissions. The heavy reliance on coal-burning for energy generation in the country is a big challenge in the process of decarbonizing the manufacturing sector. Although India still relies heavily on coal to meet the energy demand, the country's focus on renewable energy generation may set it on the low carbon emission path. The country has a goal of generating 175 GW of power through renewable sources by 2022 which comprises 100 GW from solar, 60 GW from wind, 10 GW from bioenergy, and 5 GW from small hydropower sources. Certain technological innovations in the field of renewable energy such as canal-top solar plants are boosting India's efforts to reduce $CO_2$ emissions in near future.

In the case of the USA, both the real GDP and renewable energy consumption variables bear a negative association with $CO_2$ emissions as reflected in Figure 1, where a downward movement in $CO_2$ emissions in the country can be observed. This finding supports the EKC hypothesis that beyond a threshold level of economic growth, any further increase in real GDP improves the environmental quality as more resources can be committed to innovating cleaner technologies and upgrading the infrastructure in manufacturing.

Earlier statistical models have estimated the elasticity values for the scale, income, and substitution effects of economic growth and trade liberalization [55–57]. These models have assumed a log-linear relationship between air pollution and income per capita and trade to GDP ratios. After estimating the elasticity values, they have added them to arrive at a net impact of growth and trade on pollution. However, as we argued in earlier sections, these models suffer from the non-adaptive behavior of the statistical relationship. The RBFNN model developed in this study helps estimate the nonlinear relationship adaptively. However, the RBFNN model does not produce equivalent elasticity values which can be added to provide a net impact.

## 7. Conclusions

The Intergovernmental Panel on Climate Change (IPCC) has warned about the catastrophic effects of global warming if the global mean temperature is not pegged at 1.5 °C above the pre-industrial level of warming by the end of the 21st century [58]. The current level of atmospheric temperature has already reached 1.2 °C above the pre-industrial level. At the Paris climate summit of 2015, about 200 countries pledged to reduce $CO_2$ emissions. In this context, the current study estimates the $CO_2$ emissions of 11 high emitting and 8 low emitting countries. The prediction of $CO_2$ emissions is done following the EKC framework, however, the study contributes to this literature by developing and using an artificial neural network model known as RBFNN.

Based on a dataset spanning 1960 to 2019, the RBFNN model can predict the $CO_2$ values of two sets of high emitting and low emitting countries with nearly 98% accuracy. The models predict based on both the traditional economic predictors as well as a novel non-economic predictor such as the political freedom index. By comparing the prediction error values of the full model with a partial model wherein renewable energy share is excluded, the simulation results show that the full model achieves higher prediction accuracy. This finding establishes with higher certainty compared to the earlier statistical models that renewable energy indeed holds the key for future $CO_2$ emission reduction, thus curbing the climate change effects.

The policy implication of this finding is that the rapidly industrializing countries such as China, India, Brazil, Iran, and Indonesia have to rethink their industrial policy and growth model. First, there is a need to innovate on cleaner technologies that would require less energy per output, and secondly, fossil fuel-based energy generation needs to be substituted with renewable energy generation. Though, both China and India have taken big strides in this direction in terms of China's push for the adoption of a circular economy model in industry and India's focus on ambitious renewable energy generation targets, they still need to allocate large investments for rapid reformation of their emission reduction plans.

This study makes two main contributions to the literature on EKC and the current climate crisis. First, the nonlinear adaptive models such as RBFNN provide accurate prediction for $CO_2$ levels of major emitting countries in the world and hence can be used in a more generalized way. Since this is an adaptive model with low complexity, it is easier to predict the future $CO_2$ emission levels accurately with less computational time. However, to implement this research idea for real policymaking, there is a need to build an emission simulation software package integrating this simulation model. This software can simulate the future emission levels of $CO_2$ and other environmental quality indicators as well such as $SO_2$, $PM_{10}$, and $NO_2$ by inputting the key predictor values to the model in real-time. Given its low computational requirement and high level of accuracy, it can equip policymakers with information for future emission paths of the countries and global emission levels. Second, as our findings show that higher renewable energy consumption can reduce $CO_2$ emissions, there should be more investments in this energy generation to replace non-renewable energy.

**Supplementary Materials:** The following supporting information can be downloaded at: https://www.mdpi.com/article/10.3390/su14095260/s1, The Excel files D1–D6.

**Author Contributions:** Conceptualization, P.R.J. and R.M.; methodology, B.M., P.R.J. and R.M.; software, B.M.; validation, B.M., P.R.J. and R.M.; formal analysis, P.R.J. and B.M.; investigation, P.R.J. and R.M.; data curation, B.M.; writing—original draft preparation, P.R.J., B.M. and R.M.; writing—review and editing, P.R.J. and B.M.; supervision, P.R.J.; project administration, P.R.J. All authors have read and agreed to the published version of the manuscript.

**Funding:** This research received funding from the National Institute of Technology Karnataka, Surathkal and the Scheme for Promotion of Academic and Research Collaboration (SPARC), Ministry of Education, Government of India [Proposal ID–302].: P-302.

**Institutional Review Board Statement:** Not applicable.

**Informed Consent Statement:** Not applicable.

**Data Availability Statement:** The data used in this study are available in the Supplementary Materials.

**Conflicts of Interest:** The authors declare no conflict of interest.

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
