# Peer review of "Estimating Long-Run Relationship between Renewable Energy Use and CO2 Emissions: A Radial Basis Function Neural Network (RBFNN) Approach"

_sustainability, doi:10.3390/su14095260_

Round 1

Reviewer 1 Report

Authors have considered reviewer comments in the manuscript and it can be published in journal.

Author Response

We thank the reviewer very much for his/her constructive comments which has helped us improve the manuscript significantly.

Reviewer 2 Report

The article now is really re-written and the most of the attention is dedicated to argumentation, analysis of obtained results and their discussion. Also language is improved. 

Article is of importance considering climate change aims

Author Response

(The authors gave the same response as above.)

Author Response

Response to Reviewer 3

The authors have improved the article very well. However, I would recommend some changes before publication as follows:

Response – We thank the reviewer very much for his/her constructive comments. We have followed them and revised the manuscript thoroughly.

“The details of the variables used and the source of the data are provided in Section 3. The RBFNN model is explained in Section 4.” Also add the line about section 5 and conclusion section 6…

Response – A line about section 5, 6 and 7 has been added in the revised manuscript.

Remove the word “selected” from the heading “Selected Literature Review.”

Response – The word is removed.

Add the following important articles related to this study in the literature:

Shehzad, K., Zaman, U., Ahmad, M., & Liu, X. (2022). Asymmetric impact of information and communication technologies on environmental quality: analyzing the role of financial development and energy consumption. Environment, Development and Sustainability, 24(2), 1761-1780.

Zeraibi, A., Ahmed, Z., Shehzad, K., Murshed, M., Nathaniel, S. P., & Mahmood, H. (2021). Revisiting the EKC hypothesis by assessing the complementarities between fiscal, monetary, and environmental development policies in China. Environmental Science and Pollution Research, 1- 16.

Zeraibi, A., Balsalobre-Lorente, D., & Shehzad, K. (2021). Testing the environmental Kuznets curve hypotheses in Chinese provinces: a nexus between regional government expenditures and environmental quality. International Journal of Environmental Research and Public Health, 18(18), 9667.

Response – These articles have been added in the ‘Literature Review’ section.

10 in PM 10 should be written in subscript (line 708)

Response – This is corrected in the revised manuscript.

Reviewer 4 Report

The Authors submitted a new, revised version of their article. Practically, all my comments have been taken into account in the new version of the article, e.g. the literature review has been extended. However – the conclusions section could be modified to a greater extent in line with my previous recommendation.

I also have comments on the current version of figure 1. Please redesign the figure, change the scale, so that you can read the CO2 emission values for countries with low emissions, for example for France or Italy. The unit of CO2 emissions also needs to be changed. Megatons should be used instead of kilotons and six zeros.

Author Response

Response to reviewer 4

We thank the reviewer very much for his/her constructive comments. We have followed them and revised the manuscript thoroughly.

Comments and Suggestions for Authors

The Authors submitted a new, revised version of their article. Practically, all my comments have been taken into account in the new version of the article, e.g. the literature review has been extended. However – the conclusions section could be modified to a greater extent in line with my previous recommendation.

Response – We have revised the Conclusion section further to include the policy implication of the results. It has been made more concise.

I also have comments on the current version of figure 1. Please redesign the figure, change the scale, so that you can read the CO2 emission values for countries with low emissions, for example for France or Italy. The unit of CO2 emissions also needs to be changed. Megatons should be used instead of kilotons and six zeros.

Response – We have redesigned the Figure 1 in which the unit of CO2 is changed from kilo ton to megaton, so the extra zeros are gotten rid of. The scale is changed so that the emission levels of low-emission countries are now readable.

This manuscript is a resubmission of an earlier submission. The following is a list of the peer review reports and author responses from that submission.

Round 1

Reviewer 1 Report

In the article, the long-term relationship between economic growth and environmental degradation (CO2 emissions) was estimated using the environmental Kuznets curve (EKC). According to the authors, there is a non-linear relationship between environmental degradation and economic predictors that requires an appropriate non-linear model. As part of the work, an adaptive and non-linear model was developed – a radial basis function neural network model (RBFNN). While reading the article, I had some doubts, which are listed below.

What dictated the selection of the analyzed countries? Why are there no countries from the European Union that plan to drastically reduce greenhouse gas emissions (Fit for 55), including Germany, which has over 2% share in the global CO2 emissions. There is also no Japan with a share of over 3%.

How do the authors explain the selection of variables? As it turns out, other variables, e.g. the product price index or even non-economic indicators, such as e.g. changes in the level of education, are also significant for environmental degradation.

The main conclusion of the paper that "higher renewable energy consumption can reduce the CO2 emissions" is rather obvious. Authors should develop conclusions based on more revealing findings. For example, reflection is lacking that other factors affect the shape of the environmental curve in backward regions, and different in highly developed regions. According to another opinion, the EKC applies only to highly developed countries. How did the selection of countries analyzed in the study affect the obtained results?

How the results are presented in the light of other works, for example Williamson C., Emission, education and politics - an empirical study of the carbon dioxide and methane environmental Kuznets curve. The Park Place Economist 2017, 2 (1), 9.

Section 2. Materials and Methods: de facto, no methods are described, only some generally known statistical data are presented.

Table 1: in the caption, complete that it is about fossil CO2 emissions, and also add the absolute values of CO2 emissions for mentioned countries.

A strange (too small) values of CO2 emissions in Figure 1. For example, for the USA in the figure, there is about 15 k-tons of CO2 per year, while in 2014 it actually amounted to about 6 billion tons.

How to understand the negative values in Figure 2?

Please provide data sources for figures 1 and 2.

Line 154: i.e. - and what's next?

Figure 4: the caption of the figure should be below it, not above it; it advisable to give separate captions for each of the figures 4(a) to 4(f).

Reviewer 2 Report

Authors are requested to consider following comments and revise the manuscript.

  • In conclusion section, please avoid general description and concentrate on major results.  
  • Please rewrite conclusion with quantitative results.
  • Please cite the reference of following sentence: "global warming as defined by 330 the Intergovernmental Panel on Climate Change (IPCC) should be pegged at 1.5º Celsius above the pre-industrial level of warming by 2030". I think it is by 2100.
  • Please review the manuscript again and avoid errors such as highlighted in page: 6 of 15.

Reviewer 3 Report

Article is dealing with important problem and the analytical methods applied are adequate and correctly described. The discussion of results is of good quality and the conclusions are argumented with the obtained data. In general, the article is interesting and will find positive responses, considering novelty of the suggested approach. Some minuses: For Figures the data sources should be indicated Fig 1 - 3.

Figure 4 is of poor quality and should be improved 

Table 4 28 % not understandable??

Reviewer 4 Report

See attached file...
